# The Comprehensive Facial Injury (CFI) Score Is an Early Predictor of the Management for Mild, Moderate and Severe Facial Trauma

**DOI:** 10.3390/jcm11123281

**Published:** 2022-06-08

**Authors:** Gabriele Canzi, Paolo Aseni, Elena De Ponti, Stefania Cimbanassi, Fabrizio Sammartano, Giorgio Novelli, Davide Sozzi

**Affiliations:** 1Maxillofacial Surgery Unit, Department of Emergency, ASST-GOM Niguarda, Niguarda Hospital, Piazza Ospedale Maggiore 3, 20162 Milan, Italy; 2Department of Emergency, ASST-GOM Niguarda, Niguarda Hospital, Piazza Ospedale Maggiore 3, 20162 Milan, Italy; paolo.aseni@ospedaleniguarda.it; 3Department of Biomedical and Clinical Sciences “L. Sacco”, University of Milano, Via Giovanni Battista Grassi 74, 20157 Milan, Italy; 4Department of Medical Physics, ASST-Monza, San Gerardo Hospital, University of Milano-Bicocca, Via Pergolesi 33, 20900 Monza, Italy; e.deponti@asst-monza.it; 5O.U. General Surgery—Trauma Team, Department of Emergency, ASST-GOM Niguarda, Niguarda Hospital, University of Milan, Piazza Ospedale Maggiore 3, 20162 Milan, Italy; stefania.cimbanassi@unimi.it (S.C.); fabriziosammartano@hotmail.it (F.S.); 6O.U. Maxillofacial Surgery, Department of Medicine and Surgery, School of Medicine, ASST-Monza, St. Gerardo Hospital, University of Milano-Bicocca, Via Pergolesi 33, 20900 Monza, Italy; g.novelli@asst-monza.it (G.N.); davide.sozzi@unimib.it (D.S.)

**Keywords:** maxillofacial injuries, facial trauma, injury severity score, facial injuries classification, Operative Time, Length of Stay

## Abstract

Identifying groups of patients with homogeneous characteristics and comparable outcomes improves clinical activity, patients’ management, and scientific research. This study aims to define mild, moderate, and severe facial trauma by validating two cut-off values of the Comprehensive Facial Injury (CFI) score and describing their foreseeable clinical needs to create a useful guide in patient management, starting from the first evaluation. The individual CFI score, overall surgical time, and length of hospitalization are calculated for a sample of 1400 facial-injured patients. Receiver Operating Characteristic (ROC) analysis and the corresponding Area Under the Curve (AUC) is tested, and a CFI score ≥4 is selected to discriminate patients undergoing surgical management under general anesthesia (Positive Predictive Value, PPV of 91.4%), while a CFI score ≥10 is selected to identify patients undergoing major surgical procedures (Negative Predictive Value, NPV of 91.7%). These results are enhanced by the consensual trend of Length of Stay outcome. The use of the CFI score allows us to distinguish between the “Mild facial trauma” with a low risk of hospitalization for surgical treatment, the “Moderate facial trauma” with a high probability of surgical treatment, and the “Severe facial trauma” that requires long-lasting surgery and hospital stay, with an increased incidence of Intensive Care Unit admission.

## 1. Introduction

The Comprehensive Facial Injury (CFI) score is a widely descriptive and simple scale used to classify the severity of facial injuries [1]. Its use is friendly for Maxillofacial surgeons but also Emergency physicians and Trauma surgeons. It has excellent informative content because it is firmly related to traditional trauma outcomes, such as Overall Surgical Time (OST) and Length of Stay (LOS), which are expressions of Trauma Center’s resource involvement in acute care [2]. A high CFI score also seems to be linked with a high incidence of associated injuries, such as head and other extra-cerebral injuries, and an increased risk of Intensive Care Unit (ICU) admission [2,3,4]. 

The categorization of injury severity helps identify groups of patients with similar characteristics and evaluate their outcomes. The previously proposed severity scores for facial injuries remain untested to summarize patients in this way [5,6,7,8,9,10,11,12,13,14,15]. 

This study aims to identify two specific CFI score cut-off values to stratify the whole sample of patients with facial injuries into homogeneous groups. For this purpose, we first define a threshold value that can distinguish patients with a high probability of hospitalization and surgical management under general anesthesia from those manageable under local anesthesia or with non-operative treatment. We also define a second CFI threshold value to identify patients with facial trauma who will need more complex and prolonged surgical procedures. 

## 2. Materials and Methods

The study considers a cohort of patients managed by a team of 5 surgeons, shared by two Level I Trauma Centers in Italy.

Patients with a diagnosis of at least one facial bone fracture or soft tissue wound were included; patients of all ages and gender are considered. Patients without thorough radiological documentation (traditional X-rays or CT) who died before undergoing maxillofacial surgery or with concomitant non-traumatic facial diseases were not included. 

The data were collected retrospectively from January 2008 to August 2016. Radiological diagnostic images stored in a specific hospital server and the photographic evaluations, systematically collected and filed, are used to calculate the value of the Comprehensive Facial Injury (CFI) score for facial trauma severity [1,2]. Each individual score was verified by comparing it with results assigned by at least two of the five members of the surgical team. 

The CFI score (Figure 1) works like a checklist, offering an anatomical and functional classification of facial injuries. The database compiler scrolls through the list and gives a partial score based on the combination of injuries reported for each patient, evaluated using radiological server and clinical/photographic documentation [2]. 

The anatomical classification divides the bony facial area into three horizontal thirds: the lower third (including the mandibular symphysis, body, angles, vertical branches, and condyles, as well as the lower dentoalveolar arch); the middle third (including the upper maxilla and upper dentoalveolar arch, zygoma, lateral and medial wall and floor of orbits, and nasal bones); and the upper third (consisting of the orbital roof and frontal bone, and involving the frontal sinus and its drainage system). 

The functional distinction results in two alternative scores for each fractured site: a lower score for compound fractures, generally needing conservative treatment or nonoperative management; and a higher score for displaced fractures, for which an open reduction and internal fixation (O.R.I.F.) are needed more frequently, leading to a longer treatment time. Each site-specific score ranges from 1 to 6 and is reciprocally proportioned according to the estimated a priori duration of the procedure required for the treatment of each fractured anatomical site.

An additional severity score is applied for comminuted fractures or with loss of substances, such as bony atrophy, which can therefore increase the complexity and duration of the corrective surgery. Multi-fragmentary Le Fort fractures are assigned to the highest Le Fort level identifiable, eventually with additional bone comminution points. Unilateral Le Fort fractures are assigned half the numeric value proposed in the CFI chart, so bilateral Le Fort fracture levels could be combined. Soft tissue injuries are evaluated separately and scores added to that obtained for the three-thirds skeletal injuries: 1 point is assigned for simple and uncomplicated lacerations; 5 points are assigned for large or complicated wounds with nerve, salivary duct, or lachrymal system involvement, loss of tissues, or retrobulbar hematoma.

The final individual score is obtained by adding together the partial results according to the patient’s injuries. 

Analyzing the clinical documentation allowed the specific measured values of three variables to be recorded, taking into account the main outcomes of the study:Duration (minutes) of surgery performed for the definitive treatment of the facial injuries.Length of Stay (days) in High Care Unit (LOS in HCU).Length of Stay (days) in Intensive Care Unit (LOS in ICU).

Low Care Units (general medicine, mental health, rehabilitation) were excluded because they were more influenced by coexisting comorbidities than injury severity.

This study follows the Declaration of Helsinki on medical protocol and ethics. Due to the retrospective nature of this study and the use of anonymous radiological data, an exemption by the local ethical committee was granted.

### Statistical Analysis

The characteristics of the examined population are summarized as absolute numbers and percentages for dichotomous variables, while continuous parameters are fully described using Mean and Median with, respectively, standard deviation (SD) and interquartile range (IQR). 

Two Receiving Operator Characteristics (ROC) analyses and the corresponding Area Under the Curve (AUC) are used to determine the capability of CFI score values to discriminate between patients who need or do not need surgical treatment (first outcome) and between patients who need interventions shorter or longer than 240 min (second outcome).

AUC > 0.800 and AUC > 0.900 are considered good and excellent, respectively, while a 95% confidence interval, not including 0.500, is considered for statistical significance. 

In order to select the values with the best combination of sensitivity and specificity, two CFI cut-off values were examined, respecting each of the two outcomes. These results have been used for patients’ stratification into three homogeneous sets: “Mild facial trauma”, defined as a patient who has a low probability of hospitalization and surgical treatment.“Moderate facial trauma” defined as patients with a high probability of hospitalization and surgical treatment under general anesthesia, with procedures shorter than 240 min.“Severe facial trauma” defined as a patient requiring a major surgical procedure and general anesthesia longer than 240 min.

The percentage of surgically treated patients, duration of surgery, LOS in HCU and ICU were compared for each of the three pre-established facial trauma groups with the Chi-Square Test or Wilcoxon sum Rank Test. *p*-value < 0.05 is then calculated for statistical significance. Correctly classified percentages and Positive (PPV) and Negative Predictive Values (NPV) were then calculated. Stata 9.0 software (Stata Corporation, College Station, TX, USA) is used for the statistical analysis.

## 3. Results

The sample counted 1406 patients: 1028 male (73.1%) and 378 women (26.9%). The average age was 39.6 (SD 20.4), range 1–98 years; 1050 patients (74.7%) were operated on under general anesthesia, and 356 patients (25.3%) underwent non-operative management or were treated on an outpatient basis.

The mean CFI score of the entire population was 5.9 (SD 4.8), and the median value was 4.5 (IQR = 3–7, range 1–40). Median CFI value in surgical patients is 5 (IQR = 4–8), while 2 (IQR = 2–3) in non-operative managed patients.

Sample size, mean, median of CFI score, overall surgical time, and LOS in HCU and ICU are shown in Table 1.

The ROC curve for CFI values, linked to the probability of undergoing surgical treatment of facial injuries under general anesthesia (Figure 2), showed a good AUC = 0.88 (95%IC = 0.86–0.90). 

The CFI value of 4 fitted the best combination of sensitivity and specificity; it was selected as the first cut-off to identify patients that will be hospitalized and managed surgically under general anesthesia. Table 2 shows statistical results using a CFI score ≥ 4 cut-off (Appendix A).

Another ROC curve was tested using a CFI score to identify patients undergoing surgical procedures lasting up to 240 min (Figure 3) and showed an excellent AUC = 0.92 (95%IC 0.89–0.95). 

The CFI value of 10 has the best combination of sensitivity and specificity and is selected as the second cut-off to identify patients with a high probability of undergoing major surgical procedures longer than 240 min. Table 3 shows statistical results using a CFI score ≥ 10 cut-off (Appendix A).

Three homogeneous populations were identified using these two cut-offs: “Mild facial trauma” (CFI score < 4), defined as a patient who has a high probability of non-operative management, with a low risk of hospitalization and surgical treatment.“Moderate facial trauma” (4 ≤ CFI < 10), defined as patients with a high probability (PPV = 91.4%) of hospitalization and surgical treatment under general anesthesia, with surgical procedures shorter than 240 min (NPV = 91.7%).“Severe facial trauma” (CFI score ≥10), defined as a patient with a high risk to of requiring major surgery under general anesthesia, longer than 240 min.

Table 4 summarizes the characteristics of these three determined sets of patients; for each group, results of LOS in HCU, and LOS in ICU variables are also reported.

Figure 4 shows the percentage of patients that are surgically treated under general anesthesia in mild, moderate, and severe facial trauma groups.

The box plots of overall surgical time, LOS in HCU, and LOS in ICU help visualize Table 4 results and the differences between mild/moderate and severe facial trauma patients (Figure 5).

## 4. Discussion

The application of scoring systems to identify groups of patients with homogeneous characteristics and comparable outcomes undoubtedly simplifies clinical management, anticipating patients’ needs, and improves scientific research, allowing comparison for different treatments. This principle can be shared by traumatology, oncology, general surgery, and medicine.

In 2019, the Comprehensive Facial Injury (CFI) score was introduced and validated as a widely descriptive and simple scale, able to graduate the clinical severity of all kinds of facial injuries [1]. It showed a high statistical significance correlated to the outcomes of traditional trauma (overall surgical time and length of hospitalization) and the incidence of associated injuries [2,3,4] and was subsequently used for many scientific researches [3,4,16,17,18]. The CFI score is distributed on a discrete interval scale, but it has not yet been evaluated for its usefulness in stratifying groups of patients with similar behavior.

Previously proposed systems for severity classification of facial traumas are few and have characteristic weaknesses. 

The craniofacial disruption score or Cooter and David Score (CDS) does not classify each fracture and does not consider the soft tissue lesions; it is, overall, a reductive, self-limiting, and arbitrary system, and its use remains marginal today [5]. 

The facial injury severity scale (FISS) has been validated with respect to an economic outcome [6]; therefore, its results are undermined by the specific surgical strategy used and by the different socio-political-economic contexts analyzed; a complete classification of the different fracture types is lacking, and high scores are required to achieve meaningful positive predictive values. However, it has been considered the best communication tool available within a multidisciplinary team for many years [11,12,13]. 

The maxillofacial injury severity score (MFISS) does not consider fractures of the upper facial third, the nose-orbit-ethmoid region, and the zygomatic-orbital area; moreover, it introduces functional variables of severity that can potentially be resolved completely with reconstructive surgery, thus nullifying its predictive power in terms of outcome [7]. 

The use of maxillofacial injury severity score (MISS) is extremely time-demanding and difficult to propose to assistant surgeons or non-specialist team members [8]. The facial fracture severity score (FFSS) does not consider differences in terms of commitment in the treatment of the specific sites analyzed and is therefore excessively simplistic in terms of information capacity [9]. 

The ZS model was inspired by the FFSS and later by the CDS; the current proposed version fails to include the involvement of the soft tissues, vascular and nervous structures, and the upper facial third [10]. 

In this paper, we define two cut-off values of the CFI score and three sets of patients with reproducible trends in the predictable management of their facial injuries. None of the aforementioned severity scores were used in this way.

“Mild facial trauma” (CFI score < 4) is considered to be a patient who is likely to undergo non-surgical treatment or who can be treated under local anesthesia, so that they could be discharged early from the emergency room towards the deferred surgical evaluation on an outpatient basis. 

Not all patients with mild facial trauma will be treated conservatively, but the PPV = 91.4% of the CFI cut-off ≥4 indicates a strong ability to anticipate the probability of requiring hospitalization and surgical treatment.

“Moderate facial trauma” (4 ≤ CFI score < 10) is a patient with a high probability of surgical treatment under general anesthesia, with a procedure shorter than 4 h; 240 min in duration is generally considered the limit beyond which surgical and anesthetic procedures become more complex. This patient should remain in the emergency room until Maxillofacial evaluation, which will likely require hospitalization for surgical treatment. Considering the patients’ general conditions, the treatment planning could foresee an early maxillofacial intervention or the execution of concurrent multi-specialistic procedures, given the reduced duration of the planned surgery. 

“Severe facial trauma” (CFI score ≥ 10) is a patient with a high probability of being treated with complex and prolonged surgical procedures lasting more than 240 min. This type of patient is most often the victim of a high-energy mechanism of injury, with associated injuries and an increased risk of admission to the ICU and higher LOS. The planned surgical treatment could be heavy, and its impact on the patient’s general condition usually leads to deferred procedures, sequencing the required surgical steps. Not all patients with severe facial trauma were treated with surgical procedures lasting longer than 240 min, but the NPV = 91.7% of the CFI cut-off ≥10 means that no patient with moderate facial trauma has undergone a surgical procedure longer than 4 h.

Applying this stratification to the entire sample, the analysis of all of the three outcomes of the study shows a consensual behavior: Moving from mild to moderate/severe facial trauma groups, the percentage of patients surgically treated striking increases (*p* < 0.0001), while severe facial trauma is characterized by a higher risk of prolonged surgical procedure, ICU admission and a greater LOS.

The main statistical limit of this study is the retrospective sampling and analysis. The CFI scoring system, principally based on the evaluation of radiological images, partially reduces this observational study’s limits.

## 5. Conclusions

With the use of the CFI score, emergency physicians, intensivists, and trauma care professionals can easily stratify patients with facial trauma into three main classes based on homogeneous characteristics and comparable clinical behaviors; this allows for simplified communication between clinicians and valuable decision-making guidance on the correct management of these patients, even in the absence of specialized facilities, on relevant issues such as the early discharge of patients from the emergency room or the foreseeable commitment of resources (estimated duration of surgery, length of hospitalization, risk of intensive care unit admission).

Scientific research has also improved from these results, comparing and studying groups of patients with similar and reproducible characteristics, facilitating the processes of benchmarking and analysis of the results between the different Centers.

## Figures and Tables

**Figure 1 jcm-11-03281-f001:**
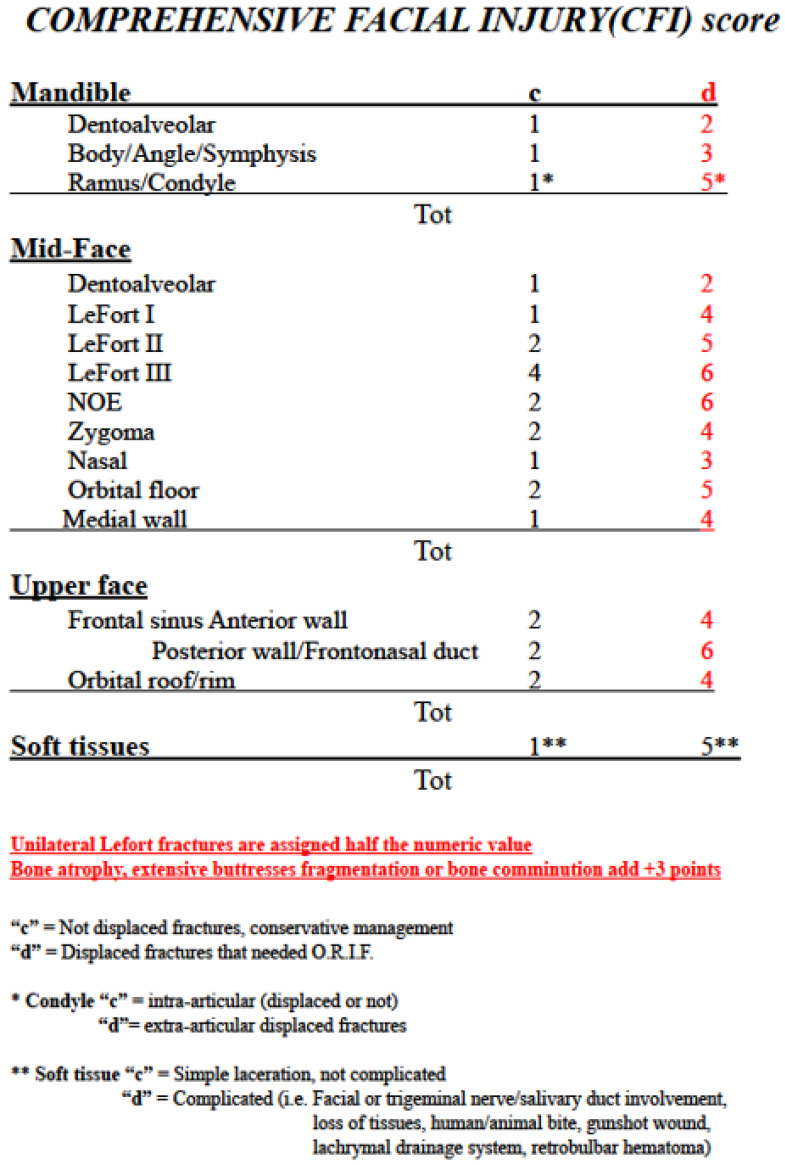
The CFI scale for estimating the severity of facial trauma. NOE = Naso-Orbito-Ethmoid fractures; ORIF = Open Reduction and Internal Fixation (Reprinted with permission from Ref. [2]. Copyright 2019 Elsevier).

**Figure 2 jcm-11-03281-f002:**
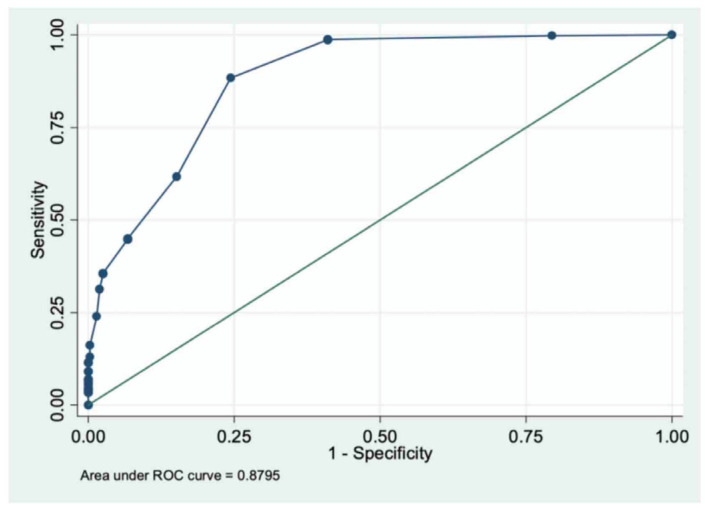
Receiver Operating Characteristic (ROC) curve for CFI score linked to the probability of undergoing surgical treatment of facial injuries. Area Under the Curve (AUC) = 0.88 (95%IC = 0.86–0.90).

**Figure 3 jcm-11-03281-f003:**
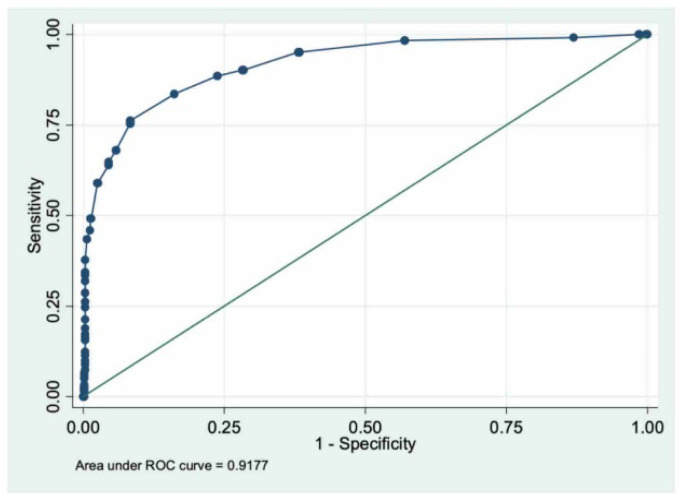
Receiver Operating Characteristic (ROC) curve for CFI linked to the probability of undergoing surgical procedures lasting up to 240 min. Area Under the Curve (AUC) = 0.92 (95%IC 0.89–0.95).

**Figure 4 jcm-11-03281-f004:**
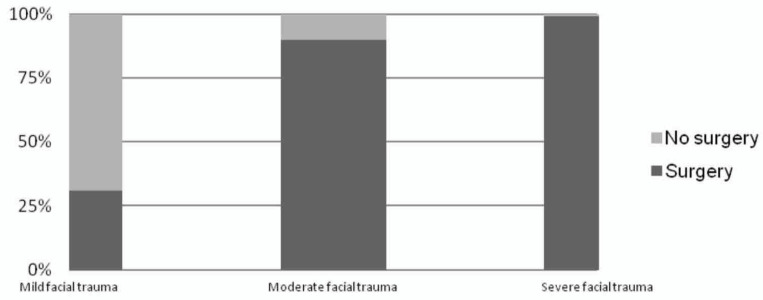
Bar chart highlighting the percentage of patients that will be treated with surgery and general anesthesia in mild, moderate, and severe stratified facial trauma patients.

**Figure 5 jcm-11-03281-f005:**
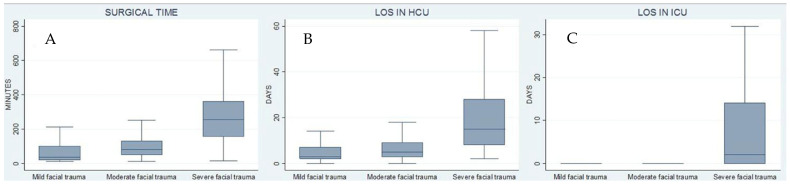
Box plots of overall surgical time (**A**), LOS in HCU (**B**), and LOS in ICU (**C**). Outside values are not represented. Box plots show the significant increase in overall surgical time (**A**) that exists between mild/moderate and severe facial trauma (according to a higher cutoff value established for CFI ≥ 10): none of the moderate classified patients reaches an overall surgical time ≥ 240 min. LOS in HCU (**B**) and LOS in ICU (**C**) results show a consensual trend, confirming the statistical goodness of the identified cutoff.

**Table 1 jcm-11-03281-t001:** Sample general characteristics and main outcomes.

Whole Sample	N° = 1406
Male (%)	1028 (73.1%)
Mean CFI score [SD]	5.9 [4.8]
Median (IQR)	4.5 (3–7)
Mean Age [SD] (years)	39.6 [20.4]
Median (IQR)	36 (23–53)
Surgery	1050 (74.7%)
Mean surgery time [SD] (minutes)	125 [121]
Median (IQR)	90 (50–160)
Mean LOS in HCU [SD] (days)	10.4 [14.9]
Median (IQR)	5 (3–11)
Mean LOS in ICU [SD] (days)	1.9 [6.7]
Median (IQR)	0 (0–0)

Sample size, mean [Standard Deviation] and median (Inter-Quartile Range) age and Comprehensive Facial Injury (CFI) score assigned. Mean [SD], and median (IQR) overall surgical time, Length of Stay (LOS) in High Care Unit (HCU), and LOS in Intensive Care Unit (ICU) for patients who were surgically treated under general anesthesia.

**Table 2 jcm-11-03281-t002:** Statistical results using Comprehensive Facial Injury (CFI) score ≥ 4 as lower cutoff.

CFI Cutoff	Sensitivity	Specificity	Accuracy	PPV	NPV
≥4	88.4% (928/1050) 86.7–90.1%	75.6% (269/356) 73.3–77.8%	85.1% 1197/1406 83.3–87.0%	91.4% 928/1015 90.0–92.9%	68.8% 269/391 66.4–71.2%

Sensitivity and specificity, accuracy, positive (PPV) and negative (NPV) predictive values (CI 95%) using CFI score ≥ 4 as a cutoff to discriminate between patients treated with non-operative-management or under local anesthesia and patients undergoing surgery under general anesthesia.

**Table 3 jcm-11-03281-t003:** Statistical results using CFI score ≥ 10 as higher cutoff.

CFI Cutoff	Sensitivity	Specificity	Accuracy	PPV	NPV
≥10	75.4% (92/122) 72.8–78.0%	91.7% (851/928) 90.0–93.4%	89.8% (943/1050) 88.0–91.6%	54.4% (92/169) 51.4–57.5%	91.7% (851/881) 95.5–97.7%

Sensitivity and specificity, accuracy, positive (PPV) and negative (NPV) predictive values (CI 95%) using CFI score ≥ 10 as a cutoff to discriminate between patients undergoing surgery and general anesthesia shorter than 240 min and patients undergoing surgery and general anesthesia longer than 240 min.

**Table 4 jcm-11-03281-t004:** Outcomes for mild, moderate, and severe facial trauma samples.

	CFI < 4 Mild Facial Trauma (N° = 391)	4 ≤ CFI < 10 Moderate Facial Trauma (N° = 845)	CFI ≥ 10 Severe Facial Trauma (N° = 170)	*p*-Value
Surgery	122/391 (31.2%)	759/845 (89.8%)	169/170 (99.4%)	<0.0001 §
Mean surgery time [SD] (minutes) Median (IQR)	60.9 [57.5] 35 (18.8–100)	97.9 [69.6] 80 (50–130)	294.0 [179.9] 255 (155–360)	<0.0001 #
Mean LOS in HCU [SD] (days) Median (IQR)	4.5 [5.0] 3 (2–5.3)	8.5 [12.0] 5 (3–9)	22.7 [23.5] 15 (8–28.5)	<0.0001 #
Mean LOS in ICU [SD] (days) Median (IQR)	0.1 [0.5] 0 (0–0)	1.3 [5.8] 0 (0–0)	8.5 [12.2] 2 (0–14)	<0.0001 #

Sample size, percentage of patients that will undergo surgery, mean [SD] and median (IQR) of overall surgical time, LOS in HCU and LOS in ICU for each set of patients defined with Comprehensive Facial Injury (CFI) score cutoff, considering only those that had surgery (§ chi-square test, # Wilcoxon sum rank test). Note the statistically significant difference in surgical patients between mild and moderate sets. The table also shows the difference in overall surgical time between moderate and severe sets of patients; this is confirmed and reinforced by the same trend of LOS in HCU and LOS in ICU results.

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
