# Peer review of "The Comprehensive Facial Injury (CFI) Score Is an Early Predictor of the Management for Mild, Moderate and Severe Facial Trauma"

_jcm, 2022, doi:10.3390/jcm11123281_

Round 1

Reviewer 1 Report

Excellent work, some minor grammatical/spelling errors but otherwise a nice read. I think it is reasonable to stratify facial trauma patients in this manner, as it can help to predict their hospital course. 

This is a well done manuscript describing the utility of a CFI score and patient injury stratification to better meet the needs of patients who are predicted to have more significant and potentially life-threatening injuries. Determining in advance what patients may need with regard to interventions can be exceptionally useful for clinicians to better treat this patient population. The readability and overall content of the paper was excellent and I enjoyed going through their statistical analysis.

Author Response

Thanks, we have corrected  the minor grammatical/spelling errors.

Many thanks for your relevant appreciation.

Reviewer 2 Report

CFI Score needs to be defined. I don't see a table or definition of how the CFI score is calculated. I had to review a different but similar article to determine the calculation. 

Author Response

Dear Reviewer, the general structure of the CFI score and the rules for its use have already been published in previous articles. We have not decided to extend this work by repeating previous publications. I attach the reference articles for convenience and I am waiting for any different decisions from you and the Editor.

Reviewer 3 Report

This is an interesting article but it must receive corrections and modifications to be accepted for publication.
Throughout the text, including the Abstract, add the meaning of the acronyms in the first citation. For example, in Abstract: ROC, AUC, PPV and NPV.
Please review Keywords as per Decs or NIH.
In the Introduction, paragraph 1, there are no citations; please add.
In Material and methods, paragraph 3, add citation of article(s) about Comprehensive Facial Injury (CFI). In this section, check if the past tense of the verbs was correctly used.
Please fully describe the title of Table 1.
In the Discussion, comment on other indexes that already exist aiming at the initial classification of cases of facial trauma and the possible difficulties that the assistant surgeon would have in classifying the severity of the trauma. Add possible articles that have already validated this CFI index.
Minor spell check of the text is required.

Author Response

The meaning of all the acronyms (CFI, ROC, AUC, PPV, NPV, OST, LOS, ICU, HCU, SD, IQR) is reported in the first citations in the text.

Keywords are reviewed as per NIH-MeSH.

Citations have been added in the Introduction paragraph as suggested.

Citations have been added in the Mat&Met paragraph as suggested. The declination of verb tenses has been checked and corrected in this section.

The title of Table 1 has been fully described.

Comments were added on other existing tools for the classification of facial trauma and the reason for possible difficulties in their use, for the assistant surgeon and team members.

Reviewer 4 Report

  1. This manuscript lacks the Figure 1 and legend text.
  2. Due to the retrospective nature of this study and the use of anonymous radiological data, the methodology needs to clearly describe about the present study selected radiological diagnostic images.
  3. Explain the CFI values of 4 and 10 fitted the best combination of sensitivity and specificity with enough detail to enable readers with access to the original data to verify the results.
  4. Please update the latest references.

Author Response

Figure 1 has been deleted and the references in the text reordered, thanks.

For each patient, as described in the Mat & Met section, we evaluated tradional x-rays and CT scans. Sensitive data was therefore obscured.

Tables 2 and 3 show, for the CFI values with their best combination, the sensitivity, specificity and accuracy of classification. Results are specified in the text. The results of all CFI values in the ROC curve are uploaded, for the proper verification by the Reviewer.

The latest available bibliographic references have been added

Round 2

Reviewer 4 Report

No comments and suggestions for Authors.

Author Response

Thank you very much.
